## [Decision Letter]

Thank you for submitting your article "Mechanistic signs of double-barreled structure of a fluoride ion channel" for consideration by *eLife*. Your article has been reviewed by three peer reviewers, one of whom, Kenton Swartz, is a member of our Board of Reviewing Editors, and the evaluation has been overseen by Gary Westbrook as the Senior Editor.

The reviewers have discussed the reviews with one another and the Reviewing Editor has drafted this decision to help you prepare a revised submission.

Summary:

This manuscript by Last and colleagues investigates the ion conduction mechanism through a novel family of fluoride selective ion channels, called Flucs. In recent years the Fluc family was characterized at the functional and structural level by the Miller lab. These proteins are dual topology dimers whose structure had raised the possibility that they contain two independent pathways for F-. However, electrophysiological recordings of single Fluc channel currents had failed to reveal any of the hallmarks of double barreled channels, such as closures to half conductance states. Actually, the location of the ion permeation pore remained somewhat elusive: four electron density peaks for F- were visible in the structure, but no continuous pore could be discerned. Therefore proof that the Fluc channels form one or two pathways was missing. In the present manuscript Last and colleagues perform a series of brilliantly designed and executed experiments to test the double barreled hypothesis. They show that 2 Phe residues in each Fluc monomer are essential for F- binding and permeation through the channel, mutation of either Phe abolishes one density peak and ion conduction. Then they use a concatameric construct where two Fluc monomers, one mutant and one WT,are tethered together. They show that in these constructs, one pore has WT properties while the other is non-conductive, showing that the Flucs are indeed double-barreled ion channels.

Overall this is an excellent manuscript and appropriate for *eLife*.

Essential revisions:

1) The manuscript is beautifully but densely written – it takes a bit of effort for the non-aficionado to navigate, and so it would be of benefit to include more detailed explanations in places. For example:

In the initial part of the results it is quite hard to discern that the structures of the single F to I mutants are new. The authors should emphasize this a bit more clearly and spell out the new information that they set out to obtain and why.

The channel records in Figure 2 should be labeled "WT homodimer + monobody" (not just "WT homodimer"), and then in the legend you could explicitly remind us that the channels don't substantially gate on their own and that closures are monobody-blocking events. (In the current version, it is simply stated that these are records in the presence of monobody.)

Figure 2—figure supplement 3 is hard to understand because it is presented before introduction of the F80I/F83I mutant.

Results section, subsection “Single-channel behavior of single-pore mutants it is stated that "the single-pore channels all recapitulate the high open probability, brief closings, and monobody block of WT". However, you haven't actually shown us any single-channel records of these single-pore channels in the absence of monobody.

In Figure 4 it is not obvious what is meant by "substate fraction" and "fractional substance conductance". Please tell us in the figure legend (not just in the text) that this means the substrate current level normalized to the full current level.

2) It is not clear to us that activity of the heteromeric concatamers in the flux assay can be taken as evidence of a two-pore assembly. While this is certainly a possible interpretation of the data, it is not the only one. Given the limited time-resolution of these fluxes, which are essentially all-or-nothing, if the mutant's effect were to reduce conductance of a common pore (or alter Po) high transport would still be expected. Similarly the inactivity of the double mutants could be explained within the context of a single-pore model.

The observation that the WT/F80I and F83I/WT concatameric channels display half the single-channel current as WT/WT is nicely consistent with the idea that one of two identical pores was eliminated. But then the F801/F83I concatamer, which in principle should also eliminate only one pore, has an even smaller conductance. Given the close proximity of the two apparent pores, this is not so surprising. But then can we really conclude that they are separate pores and not co-joined pores to start with? The authors are careful about the conclusion in their title but not in the abstract. The alternative interpretation really should be discussed, and the wording in the abstract softened. Indeed, even if there are two separate pores, if they can't be gated ("shot") independently, it's not really a "double-barreled" architecture, so the abstract is a bit misleading. Even without concluding that Flucs are bona fide double-barreled channels, the results still contribute importantly to our understanding of these channels.

We suggest you tone down your statements in the abstract and at the beginning of the paragraph "The flux behavior of the Phe-box mutants in the fused-domain channels meet all expectations of two-pore assembly[…]" and then discuss alternate interpretations in the discussion.

---

## [Author Response]

*Essential revisions:*

*1) The manuscript is beautifully but densely written – it takes a bit of effort for the non-aficionado to navigate, and so it would be of benefit to include more detailed explanations in places.*

We acknowledge and agree with the reviewer's general point and have endeavored to expand in several places to make the reader's journey through the paper a bit easier.

*For example:*

*In the initial part of the results it is quite hard to discern that the structures of the single F to I mutants are new. The authors should emphasize this a bit more clearly and spell out the new information that they set out to obtain and why.*

This is now reworded to indicate that Figure 1 shows newly solved structures (but we see no reason to blow horns), and to indicate the idea to be tested by the structures and flux measurements.

*The channel records in Figure 2 should be labeled "WT homodimer + monobody" (not just "WT homodimer"), and then in the legend you could explicitly remind us that the channels don't substantially gate on their own and that closures are monobody-blocking events. (In the current version, it is simply stated that these are records in the presence of monobody.)*

OK, done

*Figure 2—figure supplement 3 is hard to understand because it is presented before introduction of the F80I/F83I mutant*

We have now removed the mutant trace from the figure and put it into an additional figure-supplement at the appropriate point in the narrative (Figure 4—figure supplement 1).

*Results section, subsection “Single-channel behavior of single-pore mutants it is stated that "the single-pore channels all recapitulate the high open probability, brief closings, and monobody block of WT". However, you haven't actually shown us any single-channel records of these single-pore channels in the absence of monobody.*

The previously established bimolecular mechanism of monobody block means that the 'unblocked' intervals in single-channel records here show the behavior of the channel in the absence of monobody. We now make this point in the figure supplement legends showing single- and double-pore channels both with and without monobody. We have also slightly changed the language in the main text.

*In Figure 4 it is not obvious what is meant by "substate fraction" and "fractional substance conductance". Please tell us in the figure legend (not just in the text) that this means the substrate current level normalized to the full current level.*

OK – label on figure and wording in legend changed as suggested

*2) It is not clear to us that activity of the heteromeric concatamers in the flux assay can be taken as evidence of a two-pore assembly. While this is certainly a possible interpretation of the data, it is not the only one. Given the limited time-resolution of these fluxes, which are essentially all-or-nothing, if the mutant's effect were to reduce conductance of a common pore (or alter Po) high transport would still be expected. Similarly the inactivity of the double mutants could be explained within the context of a single-pore model.*

*The observation that the WT/F80I and F83I/WT concatameric channels display half the single-channel current as WT/WT is nicely consistent with the idea that one of two identical pores was eliminated. But then the F801/F83I concatamer, which in principle should also eliminate only one pore, has an even smaller conductance. Given the close proximity of the two apparent pores, this is not so surprising. But then can we really conclude that they are separate pores and not co-joined pores to start with?*

We disagree that there is insufficient evidence to conclude two-pore assembly. Taken in isolation and without knowledge of the protein's structure, the functional behavior would, as the reviewer asserts, be difficult to interpret in terms of pore-disposition. However, our experimental design for the functional assays explicitly relies upon the crystallographic results, which show the two pairs of bound F^-^ ions, isolated along two well-separated polar tracks. The structure alone implies the two-pore construction and makes obvious predictions for the flux behavior, which are herein borne out by the functional assays. It is the combination of structural data and functional results that tells the complete story and nails down a compelling case in favor of two-pore construction.

This point may not have been highlighted as it should have been in the original text, so we have added explanations to both the results and Discussion section to try and make this crucial point more clearly.

We do actually agree with the reviewer that the pores are likely to be "co-joined" – in the vestibules. We were too terse in our original text and neglected to point out that we consider the pores, though separate in the depths of the protein, are probably accessed by ions from the wide vestibules, where the monobody blockers bind. We now expand the narrative to discuss this picture and the evidence for it (last paragraph of Discussion), and have added a figure (Figure 5) to make this idea explicit.

*The authors are careful about the conclusion in their title but not in the abstract. The alternative interpretation really should be discussed, and the wording in the abstract softened. Indeed, even if there are two separate pores, if they can't be gated ("shot") independently, it's not really a "double-barreled" architecture, so the abstract is a bit misleading. Even without concluding that Flucs are bona fide double-barreled channels, the results still contribute importantly to our understanding of these channels.*

We are reluctant to enter into a semantic tussle about the meaning of "barrel." We do understand that to the reviewer "double barreled" carries the connotation that the pores are completely separate along their entire transmembrane trajectories from bulk solution to bulk solution, as with CLC channels. While we do not have quite the same interpretation of the metaphor, we suggest a compromise, now embodied in the revised text – replacement of some references to double barrels by less connotative phrases such as "dual-pathway", but not total extirpation of the offending phrase.

*We suggest you tone down your statements in the abstract and at the beginning of the paragraph "The flux behavior of the Phe-box mutants in the fused-domain channels meet all expectations of two-pore assembly…" and then discuss alternate interpretations in the discussion.*

We do not consider the statements put forward in the abstract and cited paragraph to be excessive. The cited paragraph seems quite bland to us, as it simply lists the results observed, which are indeed consistent with the 2-pore assembly previously predicted by the crystal structure. We hope that our expanded results and discussion, along with the additional Figure 5, clarifies this issue and addresses the reviewer's valid point about co-joining the pores on the two ends of the channel. We have softened the abstract a bit, however, according to the reviewer's suggestion.